# Voiding Dysfunction Due to Urethral Sphincter Dysfunction Might Be an Early Neurological Presentation of Central Nervous System Disorders in Aged Patients

**DOI:** 10.3390/jpm13040693

**Published:** 2023-04-20

**Authors:** Guan-Ru Ho, Chiao-Wen Wei, Hann-Chorng Kuo

**Affiliations:** 1Department of Medical Education, Taipei Tzu Chi Hospital, Buddhist Tzu Chi Medical Foundation, New Taipei City 231405, Taiwan; 2Department of Medical Education, Kaohsiung Chang Gung Memorial Hospital, Kaohsiung City 83301, Taiwan; 3Department of Urology, Hualien Tzu Chi Hospital, Buddhist Tzu Chi Medical Foundation and Tzu Chi University, Hualien 97004, Taiwan

**Keywords:** urethral sphincter, central nervous system disease, cerebrovascular accident, Parkinson’s disease, dementia, urodynamics

## Abstract

Purpose: To investigate the incidence of central nervous system (CNS) diseases in adult patients with voiding dysfunction and videourodynamics (VUDS) proven urethral sphincter dysfunction. Methods: This retrospective analysis reviewed the medical charts of patients aged > 60 years who underwent VUDS for non-prostatic voiding dysfunction from 2006 to 2021. A chart review was performed to search for the occurrence and treatment of CNS diseases after the VUDS examination up to 2022. The diagnosis of CNS disease, such as cerebrovascular accidents (CVA), Parkinson’s disease (PD), and dementia, by neurologists was also retrieved from the charts. Based on the VUDS findings, patients were divided into the following subgroups: dysfunctional voiding (DV), poor relaxation of the external sphincter (PRES), and hypersensitive bladder (HSB) and coordinated sphincter subgroups. The incidence of CVA, PD, and dementia in each subgroup was recorded and compared among them using one-way analysis of variance (ANOVA). Results: A total of 306 patients were included. VUDS examinations revealed DV in 87 patients, PRES in 108, and HSB in 111. Among them, 36 (11.8%) patients had CNS disease, including CVA in 23 (7.5%), PD in 4 (1.3%), and dementia in 9 (2.9%). Among the three subgroups, the DV group had the highest incidence rate of CNS disease (*n* = 16, 18.4%), followed by PRES (*n* = 12, 11.1%) and HSB (*n* = 8, 7.2%). However, no significant difference was noted in the incidence of CNS disease across the three subgroups. Nevertheless, the incidence of CNS disease was higher in patients with DV and PRES than that in the general population. Conclusions: The incidence of CNS diseases was high in patients aged > 60 years with voiding dysfunction due to urethral sphincter dysfunction. Patients with VUDS-confirmed DV had the highest incidence of CNS disease among the three subgroups.

## 1. Introduction

Dysfunctional voiding (DV) is frequently experienced by middle-aged and elderly patients. It is defined as voiding with an intermittent and/or fluctuating flow rate due to involuntary intermittent contractions of the periurethral striated or levator ani muscles, which is characterized by increased external sphincter activity or poor relaxation of the external urethral sphincter (PRES) during the voiding phase, in neurologically normal people [1,2,3]. DV is a complex condition that is not fully understood and often goes undiagnosed. It can have various underlying causes and result in dyssynergic sphincter activity in the absence of a clear neurological explanation. The precise diagnosis of DV is typically determined through videourodynamic study (VUDS) [4], which is considered an invasive diagnostic procedure. Effective treatment options for DV may include biofeedback pelvic floor muscle exercise [5,6] and injections of botulinum toxin A into the external urethral sphincter [7]. Previous studies had classified DV into two subtypes (i.e., DV and PRES), despite the fact that their treatment may be similar [4].

Previous reports indicated that central nervous system (CNS) disorders have early urological features. In one study of multiple sclerosis, the most common urinary symptom was urinary urgency (62%), followed by frequency (50.4%), urge incontinence (44.7%), and nocturia (33%) [8]. Regarding Parkinson’s disease (PD), patients with early PD may have storage lower urinary tract symptoms (LUTS) and mild motor symptoms. They usually report nocturia, urgency, and difficulty in voiding, and present with detrusor overactivity (DO) on urodynamics examination [9]. In patient with dementia with Lewy bodies, LUTS are more likely to be an early feature of disease [10]. In addition, recent studies have shown that LUTS can contribute to the development of cardiovascular events through various mechanisms such as autonomic nervous system dysfunction, mood disorders, or the negative side effects of medications [11], and LUTS have been identified as a significant predictor of acute cardiovascular events [12]. Cardiovascular disease is a known risk factor for dementia, as well as stroke [13,14]. However, it is not clear if lower urinary tract dysfunction is an early warning sign of CNS disorders.

Based on our previous clinical experience of VUDS in patients with LUTS, we noticed some patients with VUDS-confirmed DV would develop CNS disorders such as PD or CVA in the next few years. In another nationwide study, Chiang et al. also found that the occurrence of LUTS was associated with an increased risk of dementia in the aging population [15]. Previous studies on stroke in Taiwan have shown that compared with younger people, the risk of stroke begins to increase significantly at the age of 45–64 years [16]. This is consistent with our clinical observation that the proportion of stroke in patients older than 60 years is increased. Therefore, we speculated that the urethral sphincter dysfunction during voiding detected through VUDS might be an early sign of a CNS lesion, which might develop some years later. For the patients with voiding dysfunction due to DV or PRES, a neurological consultation might help to diagnose the CNS disease early and treat them earlier to achieve better therapeutic outcomes.

## 2. Materials and Methods

This study retrospectively reviewed patients aged > 60 years with voiding dysfunction who underwent a VUDS from January 2006 to December 2021 in our hospital. The inclusion criteria were patients who were neurologically intact but had voiding dysfunction other than anatomical bladder outlet obstruction (BOO), including bladder neck dysfunction, prostatic obstruction, and urethral stricture, or detrusor underactivity. Patients with previous spinal cord injury or overt CNS diseases, such as cerebrovascular accidents (CVAs) or cerebral degenerative diseases, were excluded. A chart review was performed to search for the occurrence and treatment of CNS diseases in the neurological department after the VUDS examination up to 2022. Data on the neurological diagnosis of CNS diseases, such as CVA, PD, and dementia, made by a neurologist were retrieved from the chart review. The incidence of CVA, PD, and dementia in each VUDS subgroup was recorded and compared among them. This study was conducted in accordance with the Declaration of Helsinki, and approved by the Ethics Committee of Hualien Tzu Chi Hospital and the Buddhist Tzu Chi Medical Foundation (protocol code IRB 105-151-B, date of approval 12/15/2016). The patient selection flow chart is shown in Figure 1.

## 3. Videourodynamic Study

The procedures and diagnostic criteria of lower urinary tract dysfunction (LUTD) in VUDS were according to the recommendations of the International Continence Society terminology [2]. The following VUDS parameters were recorded: cystometric bladder capacity (CBC), intravesical pressure (Pves), voiding detrusor pressure (Pdet), maximum flow rate (Qmax), corrected maximum flow rate (cQmax, defined as Qmax/CBC^1/2^), voided volume (Vol), post-void residual volume (PVR), voiding efficacy (VE, defined as vol/CBC), bladder contractility index (BCI, defined as Pdet Qmax + 5 × Qmax), and female bladder outlet obstruction index (BOOIf, defined as Pdet—2.2 × Qmax).

According to the VUDS findings, we categorized the patients with voiding dysfunction into the following three subgroups: patients with DV, patients with PRES, and patients with a hypersensitive bladder (HSB) and a coordinated urethral sphincter. The mainstay diagnostic finding of DV and PRES was the narrow urethral sphincter during voiding, indicating a discoordinated external sphincter during voiding. VUDS-confirmed DV was defined as having a high voiding Pdet (>40 cmH_2_O in men and >30 cmH_2_O in women) with a low Qmax (<15 mL/s in men and <20 mL/s in women) and a narrow urethral sphincter in the voiding phase of VUDS. PRES was defined as having a low or normal voiding pressure (≤40 cmH_2_O in men and ≤30 cmH_2_O in women) with a narrow mid-urethra or distal urethra. Patients with an HSB and coordinated urethral sphincter usually presented with an early fullness and urgency sensation during the filling phase, and a low detrusor contractility and a low Qmax, with or without an increase in PVR (Figure 2).

## 4. Statistical Analyses

Statistical analyses were performed using the SPSS software package (SPSS Inc., version 25, Chicago, IL, USA). Continuous variables are presented as mean ± standard deviation. Student’s t-test was used to analyze continuous variables in comparison between patients with and without CNS lesions. One-way analysis of variance (ANOVA) was performed followed by post hoc least significant difference calculation, which was used for the post hoc statistical multiple comparisons of VUDS parameters in the three subgroups. The chi-squared test was used for the comparison of categorical variables. In all cases, a two-tailed *p* value < 0.05 was considered statistically significant.

## 5. Results

The total number of VUDS for adults with both storage and voiding LUTS from January 2006 to December 2021 in our hospital was 763. There were 507 patients aged > 60 years. Among them, 201 patients with spinal cord injury or overt CNS diseases were excluded; therefore, only 306 (40.1%) patients were included in this study. The participants’ characteristics are shown in Table 1. The mean age of the patients at the time of the first VUDS study was 69.7 ± 6.9 years. There were 69 (22.5%) male patients. Among the 306 patients, DV was diagnosed in 87, PRES in 108, and HSB in 111. After a median follow-up period of 5.7 (range: 1–15) years, 36 (11.8%) patients had CNS disease, including CVA in 23 (7.5%), PD in 4 (1.3%), and dementia in 9 (2.9%). Among the three subgroups, patients with DV had the highest incidence of CNS disease (*n* = 16, 18.4%), followed by PRES (*n* = 12, 11.1%) and HSB (*n* = 8, 7.2%). There was no significant difference in the incidence of hypertension, diabetes mellitus, dyslipidemia, or constipation or high body mass index among the three groups.

Table 2 lists the distribution of different CNS diseases in the three subgroups of patients with voiding dysfunction. No significant difference was noted in the occurrence rate of CNS diseases among the three subgroups (*p* = 0.091). However, the incidence of CVA was relatively higher in patients with DV (12.6%) than in those with PRES (7.4%) or HSB (3.6%).

According to the VUDS findings, we divided the patients with voiding dysfunction into three subgroups, i.e., the DV, PRES, and HSB groups. Table 3 lists the baseline VUDS parameters of the three subgroups. Patients with DV had a significantly higher rate of detrusor overactivity (DO), higher Pdet, lower Qmax, smaller voided volume, and higher BOOI than the other groups. Patients with PRES normally had a low Pdet, smaller voided volume, lower BCI, and low BOOI. These VUDS parameters might not be useful to differentiate the groups from each other. Using the voiding cystourethrography during the voiding phase, however, the three voiding dysfunction subtypes can be clearly differentiated.

We also examined the incidence of DO in patients with CNS diseases. In the DV subgroup, a total of eight (8/16, 50%) patients had DO; of these patients with DO, five had CVA (5/11, 45.5%), one had dementia (1/2, 50%), and two had PD (2/3, 66.7%). In PRES, four patients had DO (4/12, 33.3%); of them, three had CVA (3/8, 37.5%) and one had PD (100%). No patients with dementia had DO. In patients with DV and PRES, there was no significant difference in the incidence of DO among the subgroups. Furthermore, no patients with CNS disease had DO in HSB.

## 6. Discussion

In this study, the incidence of CNS diseases was high in patients aged > 60 years with urethral sphincter discoordination during voiding. The incidence of CNS diseases in patients with VUDS-confirmed DV had the highest incidence of CVA among the three voiding dysfunction subgroups. This result suggests that a discoordinated urethral sphincter detected in VUDS in patients with voiding dysfunction might have association with an abnormal link in the micturition reflex network in the brain and an abnormal endothelial function or possible cerebrovascular diseases. An early detailed neurological examination or referral to a neurologist for a further investigation of the CNS disorder is necessary and early treatment to prevent CVA or PD might be given.

This study investigated whether urethral sphincter dysfunction could be an early neurological presentation of CNS disorders in aged patients with clinically diagnosed voiding dysfunction. We reviewed the medical records of patients with voiding dysfunction who had abnormal VUDS findings in the past 15 years. According to the results of the VUDS, we divided the patients with voiding dysfunction into the DV, PRES, and HSB subgroups. The cause of voiding dysfunction in patients with HSB is the increased bladder sensation during bladder filling, and patients usually have a low detrusor contractility because the detrusor muscle has not been fully stretched, leading to a low Qmax, and difficulty in urination or incomplete bladder emptying associated with urinary frequency and nocturia [17]. However, patients with HSB have normal coordinated urethral sphincter relaxation during voiding. In contrast to HSB, the pathophysiology of DV and PRES usually involves an abnormality in the CNS or peripheral pathway. Therefore, we considered using patients with HSB as a comparative arm to investigate whether the incidence of CNS diseases in aged patients with voiding dysfunction due to DV or PRES might be higher than that in patients with HSB. Although there was no significant difference in the clinical characteristics between the DV, PRES, and HSB subgroups, we found that the incidence of CVA was relatively higher in patients with DV (12.6%), followed by patients with PRES (7.4%) and HSB (3.6%). However, the incidence of dementia and PD did not differ across the three subgroups.

In Taiwan, the incidence of CVA in patients aged 36 years or older was 0.3% [18]. In another study in Taiwan, the age-adjusted incidence of all strokes ranged from 0.21% to 0.25%, according to different years [19]. Globally, the incidence of CVA in patients aged > 55 years old ranged from 0.42% to 0.65% [20]. In Taiwan, the incidence of PD was 0.028% in adult patients aged > 40 years [21]. In another study in Taiwan, the age–sex-standardized incidence of PD ranged from 0.033% to 0.036%, according to different years [22]. In North America, the incidence of PD among those older than 65 years old ranged from 0.11% to 0.21% [23]. Although the distribution of patients with CNS diseases in this study was not significantly different across the DV, PRES, and HSB subgroups, the incidence of CVA and PD in the population with urethral sphincter dysfunction was relatively higher than that of the general population.

In the brain, many neuron populations are involved in multiple pathways carrying information between the brain and the spinal cord, in order to control the bladder, the urethra, and the urethral sphincter [24]. The normal urination of humans with intact neurological function is characterized by detrusor contraction and relaxation of the urethral sphincter and pelvic floor muscles. Micturition is triggered by the release of tonic inhibition from the supra-pontine centers and the release of the trigger signal from the pontine micturition center. The relaxation of the urethral sphincter and pelvic floor muscles via the pudendal and hypogastric nerve innervations is important in the initiation of the micturition cycle. Adequate relaxation of these muscles results in sustained bladder detrusor contractions, whereas the interruption of urethral sphincter relaxation during micturition will inhibit sustained detrusor contraction [25]. Patients with chronic brain disorders might have an error link in the voiding phase, which results in urethral sphincter dyssynergia and poor external urethral sphincter and pelvic floor muscle relaxation. In humans, damage to the supra-pontine circuitry caused by lesions in the anterior cerebral region or degeneration of dopaminergic neurons in Parkinson’s disease can lead to a loss of tonic inhibitory control over the pontine micturition center, leading to reduced bladder capacity and detrusor overactivity [24]. Moreover, studies on the effects of pharmacological interventions in animals with supra-pontine lesions caused by decerebration or cerebral infarction have shown that such lesions can result in increased activity of excitatory mechanisms involving the NMDA-glutamate and D2-dopamine receptors, while the activity of inhibitory mechanisms involving the NMDA-glutamate and M1-muscarinic receptors in the brain is reduced [26]. These findings suggest that the effects of supra-pontine lesions on bladder activity are partially mediated by changes in synaptic transmission in the pontine micturition center [27].

DV is caused by the discoordination between the pelvic floor–external sphincter complex and the detrusor muscle [28]. Patients with stroke may have detrusor dysfunction (hyperreflexia or hyporeflexia) [29]. Another study revealed that patients with PD may have DO and sphincter bradykinesia with delayed relaxation of the urethral sphincter and pelvic muscles when attempting to void [30,31]. The DO in PD is caused by neurodegenerative changes within the basal ganglia, with a consequent decrease in dopaminergic function, resulting in the disinhibition of the micturition reflex [32]. Impairment of the age-related neurovascular unit including neurons, glial cells, vascular cells, and the basal lamina matrix will facilitate the pathophysiological process of Alzheimer’s disease as well as PD, and result in the vulnerability of the aged neurovascular unit to developing ischemic stroke [33]. This pathophysiology might explain why the incidence of CVA and PD in DV patients in the following years is higher in aged patients with an initially intact neurological presentation who had VUDS-confirmed DV than that in the general population.

According to the Global Burden of Disease, Injury, and Risk Factors study (GDB 2010), stroke is listed as the second most common cause of death after ischemic heart disease, with a 26% increase in stroke mortality since 1990 [34]. Although stroke-related mortality has gradually declined in recent years, the number of lost disability-adjusted life years has continued to increase [35]. Stroke risk factors include atrial fibrillation, hypertension, revascularization, hyperlipidemia, antiplatelet drugs, smoking, diet, and physical inactivity [36]. Taking control of hypertension as an example, achieving a systolic blood pressure of <130 mm Hg, compared with 130 to 139 mm Hg, appears to provide additional stroke protection in people with risk factors [37]. If dysfunctional voiding can be regarded as a sign of cerebrovascular disease, evaluation can be performed and can achieve the purpose of early diagnosis and early treatment.

To the best of our knowledge, this is the first study to investigate the relationship between voiding dysfunction and an early neurological presentation of CNS diseases in aged patients. The strength of this study is that we utilized VUDS examination for the diagnosis of urethral sphincter dysfunction. A VUDS is a test that combines voiding pressure, urine flow, urethral sphincter electromyography, and imaging studies during the bladder storage and voiding phases, which can be of great help in differentiating DV from other LUTDs, such as PRES or anatomical BOO [38]. The pressure flow studies and concomitant voiding cystourethrography in one VUDS work provide a clear view of the status of the bladder, bladder outlet, and upper urinary tract during the bladder storage and voiding phases [39]. A secondary strength is that the patients were longitudinally followed up for a median period of 5.7 (range: 1–15) years after the VUDS examination. This long follow-up time enables us to detect any neurological disorder occurring.

This study had several limitations. First, it was a retrospective study. Therefore, some patients were only followed up in the urology department, leading to an underestimation of the number of patients with CNS diseases in the following years. Second, although patients with DV had a higher incidence of CVA and PD than the general population, there was no significant difference in this incidence across the three subgroups. This might be due to the small number of patients included in this study. Third, because the pathophysiology of DV remains unclear, there is no direct evidence to demonstrate that the early stage of CVA and PD might have a direct link to DV or PRES. Fourth, some patients were followed up in a short period, so it is possible that some of them might develop a CNS disorder in the longer follow-up, resulting in false negative cases. Finally, the ethnic group in this study is mainly the Asian Taiwanese population; thus, extrapolation to other ethnic groups cannot be determined.

## 7. Conclusions

The incidence of CNS diseases during the follow-up period was high in patients aged > 60 years who did not have BOO-related voiding dysfunction. Patients with VUDS-confirmed DV had the highest incidence of CVA and PD among the three voiding dysfunction subgroups, and the incidence was higher than that of the general population. In elderly patients with no neurological disorders, presentation with voiding dysfunction due to urethral sphincter discoordination might be an early neurological presentation of CNS diseases and they may need a detailed neurological examination to detect potential CNS diseases.

## Figures and Tables

**Figure 1 jpm-13-00693-f001:**
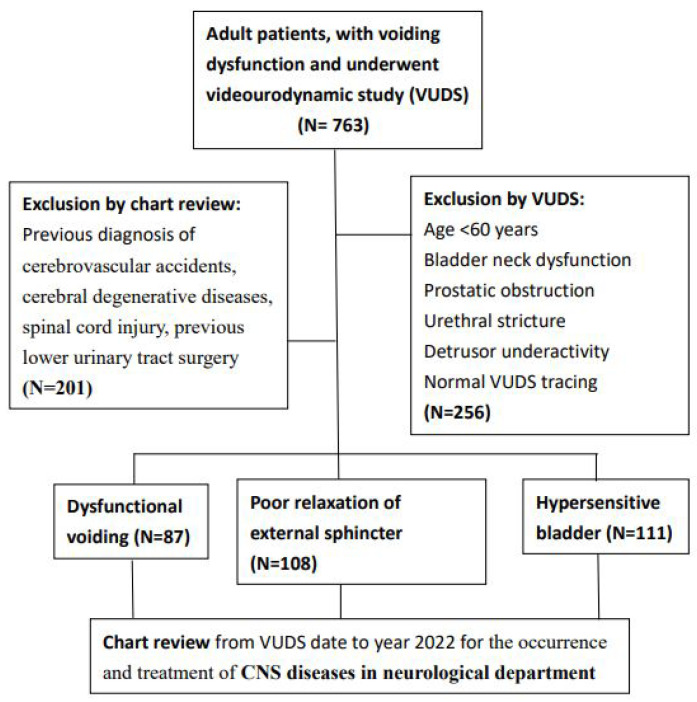
The patient selection flow chart.

**Figure 2 jpm-13-00693-f002:**
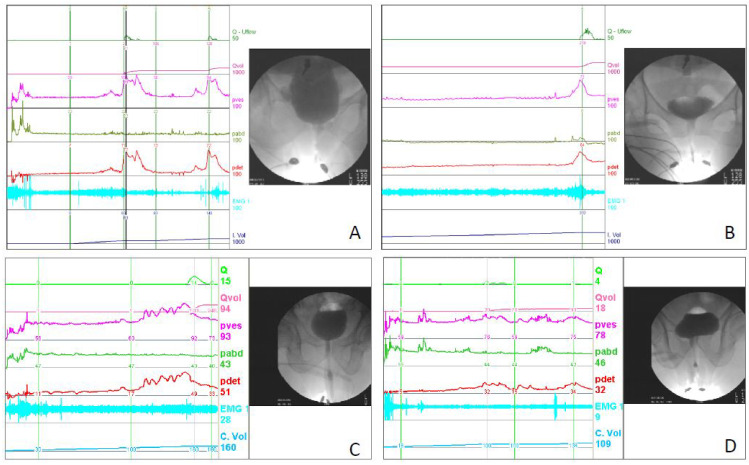
The videourodynamic tracings of dysfunctional voiding: (**A**) dysfunctional voiding in a woman with cerebrovascular accident, (**B**) detrusor overactivity and dysfunctional voiding in a woman with mild dementia, (**C**) poor external sphincter relaxation in a man with Parkinson’s disease, and (**D**) detrusor overactivity with low detrusor contractility and coordinated external sphincter in a woman with senile dementia. Abbreviations: Q: flow rate, Qvol: voided volume, Pves: intravesicle pressure, Pabd: intra-abdominal pressure, Pdet: voiding detrusor pressure, EMG: electromyogram, C. Vol: cumulated infused volume.

**Table 1 jpm-13-00693-t001:** Patients’ demographics.

	All (*n* = 306)	DV (*n* = 87)	PRES (*n* = 108)	HSB (*n* = 111)	*p* Value
Age	69.7 ± 6.9	71.9 ± 7.6	69.9 ± 6.7	67.8 ± 5.9	*p* < 0.001
Male gender	69 (22.5%)	28 (32.2%)	34 (31.5%)	7 (6.3%)	*p* < 0.001
Hypertension	154 (50.3%)	45 (51.7%)	62 (57.4%)	47 (42.3%)	*p* = 0.061
Diabetes	86 (28.1%)	31 (35.6%)	31 (28.7%)	24 (21.6%)	*p* = 0.113
Dyslipidemia	92 (30.1%)	24 (27.6%)	32 (29.6%)	36 (32.4%)	*p* = 0.771
Constipation	79 (25.8%)	24 (27.6%)	34 (31.5%)	21 (18.9%)	*p* = 0.087
BMI > 24	97 (31.7%)	32 (36.8%)	33 (30.6%)	32 (28.8%)	*p* = 0.382
Urodynamic DO	90 (29.4%)	56 (66.7%)	25 (23.1%)	9 (8.1%)	*p* < 0.001

DV: dysfunctional voiding, PRES: poor relaxation of external sphincter, HSB: hypersensitive bladder, BMI: body mass index.

**Table 2 jpm-13-00693-t002:** The distribution of patients with central nervous system diseases stratified by different voiding dysfunction subtypes.

	DV (*n* = 87)	PRES (*n* = 108)	HSB (*n* = 111)	Total (*n* = 306)
CNS diseases	16 (18.4%)	12 (11.1%)	8 (7.2%)	36 (11.8%)
CVA	11 (12.6%)	8 (7.4%)	4 (3.6%)	23 (7.5%)
Dementia	2 (2.3%)	3 (2.8%)	4 (3.6%)	9 (2.9%)
PD	3 (3.5%)	1 (1.0%)	0	4 (1.3%)

CVA: cerebrovascular accident, PD: Parkinson’s disease.

**Table 3 jpm-13-00693-t003:** The baseline videourodynamic parameters among the three groups of patients with voiding dysfunction.

	DV (*n* = 16)	PRES (*n* = 12)	HSB (*n* = 8)	Total (*n* = 36)	*p*	Post Hoc
PVR (mL)	211 ± 222	163 ± 191	100 ± 80.2	171 ± 189	0.405	
FSF (mL)	141 ± 59.2	151 ± 64.0	147 ± 66.4	146 ± 60.8	0.909	
FS (mL)	207 ± 109	279 ± 121	237 ± 66.4	238 ± 108	0.220	
US (mL)	234 ± 126	338 ± 142	325 ± 107	289 ± 134	0.084	
Compliance	54.3 ± 64.8	65.4 ± 53.2	118 ± 118	72.2 ± 78.3	0.159	
Pdet (cmH_2_O)	35.4 ± 19.6	17.5 ± 10.3	18.8 ± 9	25.7 ± 17.0	0.006	DV vs. PRES
DO	8 (50.0%)	4 (33.3%)	0	12 (33.3%)	0.064	
Qmax (mL/s)	6.91 ± 4.68	5.67 ± 4.31	14 ± 9.44	8.1 ± 6.6	0.010	DV/PRES
Volume (mL)	167 ± 130	168 ± 163	312 ± 114	200 ± 148	0.047	
CBC (mL)	379 ± 185	332 ± 170	412 ± 77.3	370 ± 161	0.538	
cQmax	0.38 ± 0.26	0.31 ± 0.22	0.69 ± 0.46	0.43 ± 0.33	0.026	PRES vs. HSB
BCI	69.9 ± 31.7	45.8 ± 28.3	88.8 ± 46.6	66.1 ± 37.2	0.030	PRES vs. HSB
VE	0.49 ± 0.37	0.52 ± 0.44	0.75 ± 0.2	0.56 ± 0.37	0.274	
BOOI	21.6 ± 21.0	6.17 ± 9.40	−9.25 ± 22.2	9.58 ± 21.6	0.002	DV vs. HSB

DV: dysfunctional voiding, PRES: poor relaxation of external sphincter, HSB: hypersensitive bladder.

## Data Availability

Data are available on request from the corresponding author.

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
