# Peer review of "Voiding Dysfunction Due to Urethral Sphincter Dysfunction Might Be an Early Neurological Presentation of Central Nervous System Disorders in Aged Patients"

_jpm, 2023, doi:10.3390/jpm13040693_

Round 1

Reviewer 1 Report

The Manuscript is about urethral sphincter dysfunction and its possible identification as early neurological presentation of a CNS disorder. The subject is very interesting, but I would like to point out something:

1) Some moderate English changes are required in order to make the manuscript more clear and easier to read.

2) The first two sentences report some observations which could better fit in the "introduction" or "discussion" section.

3) When the patients were divided into groups, what cutoffs were used?

4) Figure 1 has not a nice resolution: is it possible to put a better figure 1?

5) A patient selection flow-chart could be added

Author Response

Reviewer #1

The Manuscript is about urethral sphincter dysfunction and its possible identification as early neurological presentation of a CNS disorder. The subject is very interesting, but I would like to point out something:

  • Some moderate English changes are required in order to make the manuscript more clear and easier to read.

Reply: Thank you for the comment. We have improved the English writing. We also added some paragraphs to enrich the content of this article (statements marked in red).

  • The first two sentences report some observations which could better fit in the "introduction" or "discussion" section.

Reply: Thank you for the comment. We have moved the two sentences in the beginning of Methods to the last paragraph of Introduction. (Lines 113-116)

3) When the patients were divided into groups, what cutoffs were used?

Reply: Thank you for the comment. We have added the cut-off values of voiding pressure and maximum flow rate in defining dysfunctional voiding and poor relaxation of external sphincter. (Line 161, 162, and lines 163-164) The mainstay diagnostic finding of DV and PRES was the narrow urethral sphincter during voiding, indicating a dis-coordinated external sphincter during voiding. (Lines 158-160)

4) Figure 1 has not a nice resolution: is it possible to put a better figure 1?

Reply: Thank you for the comment. We have replaced figure 1 with a higher resolution in figure 2. (Line 168)

5) A patient selection flow-chart could be added

Reply: Thank you for the comment. We have added a patient selection flow chart (Figure 1) in the Method section. (Line 143)

Reviewer 2 Report

Welcome

Please explain to me what is new in your study?

What impact it has on the clinical practice?

This is just a statistic study.

Author Response

Reviewer #2

Please explain to me what is new in your study?

Reply: Thank you for the comment. This is the study to investigate the relationship between urethral sphincter dysfunction via VUDS and an early neurological presentation of CNS diseases in the aged patients. Although the small number of patients included in this study, patients with urethral sphincter dysfunction had a higher incidence of CVA and PD than the general population.

To the best of our knowledge, this is the first study to investigate the relationship between voiding dysfunction and an early neurological presentation of CNS diseases in the aged patients. (Lines 314-316) Previous reports indicated patients with central nervous system disorders such as CVA, PD, dementia, and intracranial lesions would have early urological features. (Lines 69-70) However, there was no report that a patient with non-obstructive voiding dysfunction especially those who have dis-coordinated urethral sphincter during voiding might have neurologic disorders years later. (Lines 109-111) Therefore, we speculated that the urethral sphincter dysfunction during voiding detected by VUDS might be an early sign of a CNS lesion, which might develop some years later. (Lines 119-120) Another strength of this study is that the patients were longitudinally followed up for a median period of 5.7 (range: 1–15) years after the VUDS examination. This long follow-up time enables us to detect any neurological disorder to occur. (Lines 323-325)

What impact it has on the clinical practice?

Reply: Thank you for the comment. This result points out that to prevent or reduce declining of brain function and CNS disorders, early screening and timely intervention, such as detailed neurological examination or non-pharmacological therapy for the elderly patients with urethral sphincter dysfunction, should be considered. It might help to early diagnose the potential CNS disease and treat them earlier to achieve better therapeutic outcomes.

The results of this study suggest that a dis-coordinated urethral sphincter detected in VUDS in patients with voiding dysfunction might have association with CNS disorders years later. (Lines 240-241) This result is important because early neurological consultation might be necessary and early treatment to prevent CVA or PD might be given. (Lines 243-245)

This is just a statistic study.

Reply: Thank you for the comment. This is not just a statistical study. To the best of our knowledge, this is the first study to investigate the relationship between voiding dysfunction and an early neurological presentation of CNS diseases in the aged patients. (Lines 314-316) In this study, we tried to count the incidence of cerebrovascular disease in patients with voiding dysfunction who underwent a VUDS examination. The result revealed that compared with the control group (HSB), DV and PRES had a higher incidence of cerebrovascular disease. Although there was no statistically significant, we still found that patients with DV had a higher incidence of stroke than the general population.

In this small cohort study, we found the incidence of CNS diseases during the follow-up period was high in patients aged >60 years who had not BOO related voiding dysfunction. Patients with VUDS confirmed DV had the highest incidence of CVA and PD among the three voiding dysfunction subgroups, and the incidence was higher than that of the general population. The elderly patients with no neurological disorder who present with voiding dysfunction due to urethral sphincter dis-coordination might be an early neurological presentation of CNS diseases and need a detailed neurological examination to detect potential CNS diseases. (Lines 340-346)

Round 2

Reviewer 1 Report

The Authors followed reviewers' suggestions and added further parts. However, I would like to point out something:

1) Introduction section seems too specialised and more like a Discussion section, especially in lines 96 - 105;

2) Minor revision: in lines 116 - 120, these two sentences seem the same, but I suggest to leave the second one;

3) In lines 309 - 311, I think that this sentence does not give further useful information to the reader. Furthermore, the Authors reported that in Taiwan hypertension si increasing, but they did not cite any reference;

4) The follow-up period range is between 1 and 15 years: are the shortest follow-up periods enough long to identify CNS disease? Because patients with short follow-up might have developed a CNS disease after their follow-up, becoming false negative cases. 

Author Response

The Authors followed reviewers' suggestions and added further parts. However, I would like to point out something:

  • Introduction section seems too specialised and more like a Discussion section, especially in lines 96 – 105

Reply: Thank you for the comment. We have deleted some irrelevant statements and moved the paragraph in Introduction to Discussion section. (Lines 253-255, and Lines 263-275)

  • Minor revision: in lines 116 - 120, these two sentences seem the same, but I suggest to leave the second one

Reply: Thank you for the comment. We have deleted the duplicated statement.

  • In lines 309 - 311, I think that this sentence does not give further useful information to the reader. Furthermore, the Authors reported that in Taiwan hypertension is increasing, but they did not cite any reference

Reply: Thank you for the comment. We have deleted this sentence, accordingly.

  • The follow-up period range is between 1 and 15 years: are the shortest follow-up periods enough long to identify CNS disease? Because patients with short follow-up might have developed a CNS disease after their follow-up, becoming false negative cases.

Reply: Thank you for the comment. We have listed this point as one of the limitations of the study. (Lines 323-325)

Round 3

Reviewer 1 Report

The Authors followed Reviewers’ suggestions and and answered their questions. The Manuscript is now more clear and interesting.